# RNA Packaging in the *Cystovirus* Bacteriophages: Dynamic Interactions during Capsid Maturation

**DOI:** 10.3390/ijms23052677

**Published:** 2022-02-28

**Authors:** Paul Gottlieb, Aleksandra Alimova

**Affiliations:** School of Medicine, The City College of New York, The City University of New York (CUNY), New York, NY 10530, USA; aalimova@med.cuny.edu

**Keywords:** procapsid, nucleocapsid, polymerase complex, assembly, replication, packaging

## Abstract

The bacteriophage family *Cystoviridae* consists of a single genus, *Cystovirus*, that is lipid-containing with three double-stranded RNA (ds-RNA) genome segments. With regard to the segmented dsRNA genome, they resemble the family *Reoviridae*. Therefore, the *Cystoviruses* have long served as a simple model for reovirus assembly. This review focuses on important developments in the study of the RNA packaging and replication mechanisms, emphasizing the structural conformations and dynamic changes during maturation of the five proteins required for viral RNA synthesis, P1, P2, P4, P7, and P8. Together these proteins constitute the procapsid/polymerase complex (PC) and nucleocapsid (NC) of the *Cystoviruses*. During viral assembly and RNA packaging, the five proteins must function in a coordinated fashion as the PC and NC undergo expansion with significant position translation. The review emphasizes this facet of the viral assembly process and speculates on areas suggestive of additional research efforts.

## 1. Introduction

In this review, we emphasize the dynamic nature of the polymerase complex (PC) protein component conformations and assembly during RNA packaging, replication, and transcription. The focus of the discussion is the motion of PC portal and framework proteins during RNA packaging, and we consider and suggest open questions where there is a need for additional research. First, a brief review of the virus, its general structure (and structural elements), and assembly are required.

The host for the viruses is the bacterium *Pseudomonas syringae pv. phaseolicola* and the first *Cystoviridae* family member, *Pseudomonas* virus phi6, was discovered and isolated from infected bean straw plants in 1973 [1]. Subsequently, additional *Cystovirus* species were found in a variety of environmental habitats demonstrating the ubiquitous nature of the virus family. In summary, these bacteriophage types offered opportunities to study RNA packaging in a dsRNA segmented genome virus but using a prokaryotic model that was relatively easy to manipulate.

## 2. The Classification of *Cystoviridae* Family

Modern ICTV classification includes seven species of the *Cystovirus* genus [2]. Until 1999, the *Pseudomonas* virus phi6 was the only species in the family *Cystoviridae* genera *Cystovirus*. *Cystoviruses* are the only dsRNA containing viruses that replicate in bacteria. The isolation of additional lipid-containing, three-segmented dsRNA bacteriophages expanded the group with additional types, some of which were closely related and others distantly related to phi6 [3]. The host range of *Cystoviruses* is governed by the structure of the attachment apparatus, as well as the specific cell surface receptor it utilizes. In the case of phi6 and its closest relatives, host infection requires binding to type IV pili [4,5]. Therefore, the phi6 group has a host range limited to *Pseudomonas syringae* and several selected mutants of *Pseudomonas pseudoalcaligenes* ERA [6]. Because phi8 and phi12 are considered distantly related to Phi6, they are able to infect *P. syringae* mutants by binding to rough lipopolysaccharide (LPS). Mutants lacking rough LPS, such as HB10Y, are not infected by phi8 and phi12. Phi12 has the ability to infect other Gram-negative bacteria containing rough LPS, such as *E. coli* JM109, but without lytic plaque formation [3]. In 2010, the *Cystoviridae* family was again expanded with the isolation of phi2954 from infected radish leaves [7]. The base sequences for many of the genes and the segment termini were similar but not identical to those of bacteriophage phi12. However, the host specificity was for the type IV pili of *Pseudomonas syringae* HB10Y rather than the rough LPS to which phi12 attaches. PhiNN was later isolated from a freshwater habitat, demonstrating that the virus group is quite diverse and widely distributed [8]. This phage utilizes the type IV pilus of *Pseudomonas* sp. B314 for attachment and phylogenetic trees of the genome segments indicate a close relationship to phi6. Notably the *Cystovirus* phiYY, which uses the opportunistic pathogen *Pseudomonas aeruginosa* strain PAO38 as a host, was successfully isolated from hospital sewage in China in 2016 [9].

## 3. Virion Structure: Protein Components and Their Viral Location

The members of the *Cystoviridae* family are all lipid-containing bacteriophages organized in a multilayered structure. The inner PC layer is composed of P1, P2, P4, and P7 proteins, where P1 is the major structural protein; P2 functions as the RNA-directed RNA polymerase (RdRp); P4 is the hexameric nucleotide triphosphorylase (NTPase) packaging motor, and P7 is an assembly and packaging cofactor. The next layer is the matrix layer, which consists of the shell protein P8 (except for phi8 virus, where the P8 is a membrane protein). The outermost layer is the cell envelope bilipid membrane layer with randomly distributed membrane proteins P6, P9, P10, and lytic endopeptidase P5 (P11) associated with the NC surface [10] and binding protein P3 (Figure 1) [11]. The functions of basic structural proteins, including their molecular weight, and the number of copies per virion are presented in Table 1.

### 3.1. Replication Cycle

The entire viral replication cycle is similar in all classified *Cystoviruses* types and is best exemplified using phi6. The schematic diagram of this cycle is shown in Figure 2. After the virions binding to the pili IV, its fusion with the bacterial outer membrane, and entry of phi6 NC into an intracellular vesicle, the P8 proteins disassemble, and RNA transcription is initiated (Figure 2A–E). Translated structural proteins P1, P2, P4, and P7 assemble as a closed and unexpanded structure (Figure 2F) [12,13]. The PC, which is responsible for RNA packaging, transcription, and genome replication, is initially assembled as a dodecahedron with recessed vertices prior to RNA packaging. P1 forms the procapsid shell [14,15]. During the replication, the PC first forms in the unexpanded state. This is followed by a sequential and stepwise expansion, with each ssRNA packaged in the following order: +s, +m, +l. Ultimately all three dsRNA segments are enclosed into the NC and surrounded by the lipoprotein envelope to constitute the mature viral particle. Crystallographic, nuclear magnetic resonance, and cryo-electron microscopy structural analysis of the components of the replication apparatus demonstrated the conformational changes and interactions required to package and replicate the RNA genome filling in gaps not entirely answered by genetic studies. The understanding of the *Cystovirus* RNA packaging and replication mechanisms established an simple model for the assembly of segmented double-stranded RNA containing viruses, in particular the *Reoviridae* [16]. While the model’s description is a consequence of genetic investigation supported by structural studies covering a period of approximately 40 years, questions still remain.Therefore, as this review addesses, the entire assembly-replication description will require additional observations.

### 3.2. Procapsid

The inner protein layer, the PC, is composed of 60 asymmetric dimers of protein P1 that form a dodecahedral T =1 shell. At each of the twelve dodecahedron faces, protein P4 hexamers (the packaging nucleoside triphosphatase, NTPase) localized at the 5-fold axis protrude from the PC producing a symmetric mismatch [17,18,19]. The RdRp P2 and protein P7 localized at the portal at the 5-fold axis of symmetry are found within the P1 shell. *Cystoviruses* package their plus sense messenger RNA (mRNA) sequentially into a preformed PC in the order s, m, and l, and the PC expands to accommodate the genome [20]. In phi6, packaging relies upon signals at the RNA 5′ end called *pac* containing about 200 nucleotides (with extensive secondary structure), while specific sequences at the 3′ end are RNA replication signals. The PC maturation step is facilitated by hinge-like movement of the entire capsid frame composed of protein P1 subunits (Figure 3) [21,22,23,24]. During the ssRNA packaging, the PC conformational morphogenesis sequential expansion reveals unique binding sites for each of the three viral RNA segments [25,26,27]. The expansion and packaging activate the P2 RdRp, which replicates the mRNA to dsRNA. Gottlieb’s group suggested that the packaged dsRNA genome segments appear to exhibit low-symmetry quasi-concentric organization based upon cryo-electron microscopy with asymmetrical reconstructions with selected class averages [28]. In the year 2019, the characterization of the phi6 genome was performed without any symmetrical restraints, and the majority of RNA follows pseudo D3 symmetry to form five concentric layers [29]. The expanded genome-filled PC is finally enclosed within the 200 P8 trimer matrix completing the assembly of the mature NC.

The *Cystoviru*s RNA portal complexes, consisting of protein elements P2, P4, and P7, are located on each of the apexes of the assembled PC. There is a potential total of twelve RNA portal complexes upon the dodecahedral PC; however, not all the sites are occupied. The dynamic interaction of these three portal proteins governs the packaging, replication, and transcription of the viral RNA genome. This intricate molecular machine can be described regarding each of the separate proteins. The assembly of the PC was achieved in *E. coli* by the expression of recombinant proteins that compose the PC [30], and purified protein constituents were self-assembled into complete PCs [31]. The PCs from both assembly methods are capable of genome packaging, replication, and transcription.

### 3.3. Nucleocapsid

The phi6 nucleocapsid (NC) surface constitutes the shell just below the envelope and has icosahedral symmetry, and consists of 200 P8 trimers, a small membrane-associated protein in a T = 13 matrix that partially covers the PC. Interestingly the individual species of the *Cystoviruses* are not entirely isomorphic, only similar in the overall design. The P8 layer is quite different among the species based on analysis of cryo-electron microscope single-particle reconstruction [22]. In phi8, the NC cover is virtually non-existent, consisting of only 60 copies of P8, while phi12 is in an intermediate category showing an incomplete P8 layer composing a T = 13 matrix [19]. Indeed, there are significant vacancies in the phi12 P8 matrix organization that is occupied by the protein P7 [32,33]. The P7 protein, its position in the PC, and potential functions are described in more detail below.

### 3.4. Outer Envelope

The outermost phospholipid bilayer is derived from the host plasma membrane and incorporates four integral proteins, P6, P9, P10, and P13, in an asymmetric arrangement. The external spike that binds the host receptor and governs the host range is protein P3. This spike protein is attached to the viral envelope and anchored to it by the integral membrane protein P6. The phi8, phi12, and phi13 outer surfaces differ radically from that of phi6 [34,35,36]. Phi8, phi12, and phi13 bind to a truncated Lipopolysaccharide (LPS) O-chain to initiate host cell infection. Only the fine structure of the phi12 surface spike was determined; it was accomplished by using cryo-electron tomography with subtomogram averaging [37,38]. The high-resolution structure of the P3 spike protein of phi6 has not been resolved yet—the small protein is randomly distributed on a bi-layer membrane, so imposing icosahedral symmetry is not possible. The structure could be resolved by either subtomogram average or by not imposing any symmetry. The phi12 spike is a protein complex of toroid-shape consisting of six globular domains with sixfold symmetry (Figure 4). Each globular domain is likely composed of one copy of protein P3a and three copies of P3c. However, as in all the *Cystovirus* types, once in proximity to the host cell outer membrane, the viral envelope fuses with that of the host with the fusion activity facilitated by the fusogenic function of protein P6 [39].

## 4. Genome Organization

The *Cystoviridae* have three dsRNA segments that are designated according to size - size L (large) contains 6374 base pairs in phi6, size M (medium) contains 4063 base pairs in phi6, and size S (small) contains 2948 base pairs in phi6 (Figure 5) [26,40]. The size of the entire genome varies from 12.7 kb for phi2954 to 15.0 kb for phi8. Each virion particle contains only one copy of each segment, and the genes are clustered into functional groups [34,41,42,43]. Each genomic segment is transcribed into a polycistronic mRNA that is designated by the corresponding lower-case letter s, m, and l. The genes encoding the structural proteins of PC are located on the L segment. While in most members of the *Cystoviridae* family, the order of the L segment genes is gene 7, gene 2, gene 4, and gene 1, in phi8, it has different order: 2, 4, 1, and 7 [3,35,44]. In phi6, the production of the RNA polymerase P2 is down-regulated to approximately 10% the level of the other three PC proteins by translational coupling to the P7 gene, as gene 2 lacks a ribosome binding site. The medium segment carries the genes encoding the host recognition complex: gene 6 and gene 3. Genes 6 and 3 are in a polar relationship with each other as mutants lacking production of one suppresses the expression of the other. Gene 3 encodes a single polypeptide in phi6, P3, whereas in phi12, phi8, and phi13, gene 3 encodes two proteins, P3a and P3c. The small segment encodes the NC protein P8, the major membrane protein P9, putative membrane morphogenetic factor P12, and the protein required for host cell lysis P5/P11. For each virus member, each of the three segments has a short, highly conserved sequence at the 5′-end (i.e., designated from the plus sense strand of the dsRNA). The highly conserved sequence is followed by a unique packaging sequence that identifies each segment and ensures that it is packaged within the PC.

## 5. Viral RNA Transcription and Packaging Signals

*Cystovirus* regulates the transcription of the three dsRNA segments by carrying identifying sequences at the 5′ ends (Figure 5). For example, in phi6, each plus-sense strand contains the 18 identical bases with one subtle exception in that *S* and *M* start with GG, while in *L,* the start is with GU [45].

Early in the viral replicative cycle, the three transcripts are synthesized more or less equally, but later in the cycle, *l* transcription is lowered [46]. Host protein YajQ binds to the P1 protein to alter the behavior of the polymerase P2 so that it favors the transcription of l with nucleotides GU during the early replication cycle [47]. In vitro transcription showes a similar temporal regulation effect with a YajQ-independent mutation in the P1 gene (L612R) and with manganese ion supplement [48]. The precise nature of how YajQ binds to P1 and regulates P2 is not understood and requires further study. However, further confounding these observations in phi6, bacteriophages phi8, phi12, and phi13 replicate independently of YajQ.

During the replication cycle, the viral transcripts are packaged as the dsRNA precursors. The packaging of the *Cystovirus* genome is dependent on cis-packaging signals located near the 5′ end of the transcripts, termed *pac.* The signals are unique to each segment, with little to no similarity among the S, M, and L sequences. In case of the phi6 virus, the packaging competency depends on 18 identical bases at the 5′ end, but not absolutely, and can tolerate few nucleotides deletion. Another conserved stretch of 12 nt CCCGGGCUACGG also located at 5′ end of all three segments: at position 83 in L, 62 in M, and 50 in S. Interestingly, while the packaging signal extends to 205 bp for L, 300 bp for M, and 250 bp for S segments, there are regions unnecessary for packaging within those segments (the deletion of nucleotides 16 to 43 did not affect the packaging competency) [49]. The structural analysis of the *pac* region of ss-RNA strands shows the low structural similarity in the first conserved 16 nt at the 5′ end. For example, in segment s, the first 10–12 nt are single-stranded; in m, a stem-loop structure is formed starting from the very 5′ end; and in l, a stem-loop structure is formed with three single-stranded nucleotides at the 5′ end. However, the internal conserved region CCCGGGCUACGG is folded as a hairpin structure with stable tetraloop UACG [50].

Mutations in the *pac* region lead to loss of the packaging ability, but this can be compensated by mutations in the P1 structural protein and by secondary mutations in the RNA [51,52]. The isolated mutants which overcome the changes in the *pac* region of M or S segments always carry the mutations in segment L, which has an Open Reading Frame (ORF) of P1 protein [52].

## 6. Maturation of the PC Mediated by Protein P1

As is shown in this section, the overall geometry of the PC framework protein, P1, governs the control of the radical conformational changes during RNA packaging, and it was the elucidation of the crystal structure of P1 from both phi6 and phi8 that demonstrated a more complete picture of the PC maturation mechanism [21,24,27,53].

Overall the shell of the PC is formed by 60 dimers of the nonequivalent P1 subunits designated P1A and P1B [21,22,27]. As shown in Figure 6c, twelve pentamers, the P1A-subunits, are centered on the 5-fold vertices, and 20 trimers of the P1B-subunit are located on the three-fold axis. The crystallographic conformation of the protein P1 PC subunits was determined for both phages phi6 and phi8, showing a similar trapezoid shape despite the very limited amino acid sequence similarity between these two P1 proteins [21]. The P1 structure was resolved by crystallography and by using an earlier cryo-EM model for non-symmetrical replacement, and the final resolution extended to 3.4 A. The trapezoidal structure is mostly α helical structure and has a variable thickness between 14 Å and 38 Å at the edges and reaching 47 Å near the center, again using the phi6 example. The sides of the protein are estimated to be approximately 91Å × 73Å, as measured for the phi6 phage. Reference to Figure 6, A denotes four distinct edges (designated I–IV), and each edge forms two different interfaces with the other P1 subunits to accommodate expansion. Major structural differences between P1A and P1B are localized in a hinge region, there four helixes are rotated to ~18° to accommodate the P1B into procapsid (Figure 6b). Each edge of the trapezoid forms two different interfaces with other P1 subunits. The 2-fold axis apposed P1B edge III forms a hinge required for PC expansion, and in this process, the planes of the subunits reorient from near perpendicular to almost co-planar in the packaged and mature nucleocapsid (Figure 6d).

Genetic evidence first suggested that P1 shape alterations govern the RNA packaging mechanism. For example, point mutations p.W103_R104delinsVA, p.R617_R618delinsAA and R385A in phi6 P1 were shown to prevent wild type s and m sequences from binding [52]. Such mutations are located on the procapsid exterior near the edge II of the P1 trapezoid (Figure 6a) and, as they are not in a clustering position, they can act independently to prevent wild-type s-segment RNA binding. Conversely, the mutations in the *pac* region of s and m RNA segments caused the compensating mutations in P1. Analysis of the location of such mutations mapped them to the positively charged P1B polypeptide segment between amino acids Cys98 to Cys155, which is believed to be possible s-segment binding site. The s-segment binding is located at a positively charged P1B tip, whereas the positively charged P1A tip is masked with a P4 hexamer and is therefore inaccessible to binding [21]. A similar idea is suggested by El Omari *et al.* for the phi8 P1 movement during PC expansion based exclusively on the analysis of PC structure. The s segment might bind at and around the 3-fold axis due to the complete masking of the 5-fold axis by the P4 hexamer. After the binding, the s overhangs the P4 ring, causing the P4 ring closure and activation of ATPase. Translocation of s RNA inside the PC causes the expansion of the PC and exposure of the binding sites for m and later l segments [53].

It was also noted the suppressors of the p.R196_R197delinsAA and p.R617_R618delinsAA mutants are clearly not in proximity to this suggested binding site, which means the long-range allosteric effects must take place in P1 when the dodecahedral PC undergoes the major conformational change during genome packaging [52].

The later structural determinations of P1 then shows a maturation process governed by the unique shape of the protein, which is responsible for the capsid plasticity that facilitates expansion to accommodate the genome. The term “plate tectonics” was used by Omari et al. [53] in the description of the PC shape-changing and assembly mechanisms of the phi8 PC and is elaborated below.

An intensive study of in vitro assembly of phi6 and phi8, performed by the D. Bamford and M. Poranen group, analyzed the PC formation from soluble proteins P1, P2, P4, and P7 (see Figure 7a). The soluble Phi6 P1 exists as monomers in solution [31], the soluble phi8 P1 as tetramers [54], P7 as dimers for both phi8 and phi6 [55,56], P2 as monomers [54,57], and P4 as hexamers [54,58] for both phi6 and phi8. The number of subunits in the oligomeric complexes in the solution is not necessarily being conserved in the assembled virion. While more indirect methods of observation, such as gel filtration and light scattering, first suggested that phi6 and phi8 assembly was initiated by utilizing tetramers of P1 [31,54,59,60] the elucidation of the x-ray, cryo-tomographic structure, and physical movement of phi8 P1 subunits compelled Omari et al. to re-evaluate the PC assembly mechanism [53]. Pentamers were noted in the crystal structure that strongly resembled the pentameric face of the PC and could be the true assembly precursors (see Figure 7b).

Therefore, Omari et al. proposed an assembly model in which P1A pentamers, shaped as collapsed 5-fold facets, are stabilized by a central P2 molecule and form higher-order trimeric assemblies. The P1B monomers recruited from the pool of soluble P1 units stabilized the PC. The P4 hexamers can be attached to the outside of the PC at the 5-fold face. El Omari et al. also suggested that the P4 further stabilizes the complex by directing the P1 pentamers into the proper curvature. From this initial 3-fold axis position, the P2 could be expected to be transported during capsid expansion toward the 5-fold axis. The initial 3-fold axis position of P2 was confirmed by multiple cryo-electron microscopy reconstructions of collapsed phi6 PC particles (see Figure 8a) [44,61,62]. The position that cryo-EM reconstructions of the entire virion of phi12 clearly indicated the mobility of P2 inside the PC during the RNA packaging shifted to a 5-fold axis directly below the P4 turret [19]. The stoichiometric data confirmed the finding, the entire P2 occupancy at a 3-fold axis was expected to be 20 in the PC, but the copy number was experimentally estimated to be between 3 and 10, suggesting that some of the three-fold axes remain unoccupied. The role of P7 in regard to its assembly function in the PC is not entirely described in this model. Nemencek et al. also described implications for PC assembly and maturation but left open the question of the P4 and P2 interactions with P1.

## 7. P4 Portals: Occupancy and Translational Motion in the PC and NC- Implications for RNA Packaging and Viral Maturation

The P4 hexamer located at the five-fold axis of the PC is the supplies the motive force required for ssRNA packaging translocation, and a brief overview of the features of the apparatus helps illustrate its dynamic function (Figure 9). The cystoviruses have 12 possible positions for the packaging portal location, and the number of sites that function at a given time to package the three genome segments remains controversial. The entire chemo-mechanical coupling was described by Mancini et al., using the crystal structure of the phi12 hexamer as the model, illustrating how ATP hydrolysis drives the viral RNA translocation through cooperative conformation changes [63]. The P4 hexamer undergoes significant dynamic behavior during the PC RNA packaging phase, and its activity alone could be subject to an entire review. However, a brief mechanistic description is presented below. We conclude this section by showing our own initial cryo-EM tomogram reconstructions that illustrated connection points of the P4 hexamer to the PC and NC.

As shown in Figure 9, the hexamer is similar in shape to a dome or funnel with an approximate diameter 95 Å and an approximate height of 55 Å. There is a central channel ranging from 21 to 25 Å in width. The flat bottom of the dome is attached to the PC, a region composed of a β sheet along with a partially disordered alpha helix (this is also true for phi12, and for phi6, the basal region is disordered). Energetic packaging of a genome into a capsid is required in many viruses, and an NTPase powered rotary micro-engine is responsible for this activity. The dsDNA viruses have elaborated multiprotein packaging complexes (terminase) [65,66] and can exert the 110 pN force necessary to translocate the dsDNA into the capsid [67], which makes it the most powerful known biomotor. The ssRNA is more flexible than the dsDNA and requires less energy to pack (the persistence length of ssRNA is l_p_ ~1–2 nm, while that of dsDNA is l_p_ ~50 nm [68]). Therefore, the packaging motor of *Cystoviruses* has a simpler structure compared to dsDNA viruses and consists of only homohexameric protein P4, with the hexameric ring functioning as both the NTPase motor and the portal [58,69]. Due to the complex secondary structure of ssRNA, the P4 motor also exhibits helicase activity. The NTPase activity of P4 protein is only exhibited in hexameric form. Notably, the ATPase action can be switched off during transcription, and the central pore of the hexameric ring can passively serve as an RNA exit [70].

Structurally, all P4 NTPase monomers can be divided into three domains: an N-terminus domain, a central ATPase core domain, and a C-terminus domain: disordered for phi6 and phi13 and slightly ordered for phi8 and phi12. The Cystoviral P4 core domain has little sequence similarity between viral species (9 to 20%) but exhibits common structural and organizational features: Rossmann type quarterly structure of β-strands connected by a-helixes [63,64]. The N-terminal domain has low similarity among species both structurally and by sequence and is located at the apical surface of the P4 protein. The stabilization factor for the hexamer is an α-helix for phi12 and disordered polypeptide chains for phi13 and phi8. Only the phi6 P4 requires the presence of nucleotides and divalent cations to form the hexamer, and presumably, the nucleotide-binding causes the hexamer stabilization [58]. The C-terminus domain is composed of 40–50 aa downstream of the ATPase core, located at the basal P4 surface facing the P1 shell is responsible for P1 binding [71,72]. For both phi6 and phi13, the C-terminal domain did not show any density of EM images and is possibly disordered.

The core domain is structurally most like the Rec-A ATPase protein. A P-loop or Walker A motif is responsible for β- and γ- phosphates of ATP binding and wraps around the polyphosphates of the nucleotide [73]. The Walker motif or phosphate-binding loop, found in many ATP and GTP utilizing enzymes, is required for NTP binding and contains the pattern G-x(4)-GK-[T/S], where x denotes any amino acid [74,75]. The phi6 Walker A motif is located at the tip of the first β-strand and conserved between cystoviruses: GATGSGKS (aa 125 to 132 for phi6). The less conserved Walker B motif AEAYDE (aa 151–156 for phi6) is required to coordinate Mg^2+^ ions. The binding of β-phosphate of the nucleotide and the NTPase activity is upregulated by Ca^2+^ and Zn^2+^ ions, while Mg^2+^ ions act as a non-competitive inhibitor [58]. The loops central to the hexamer drive the RNA packaging by pivoting to an “up or down” position. The alpha helix L2 loop and P loop (see Figure 9c,d) move downward following ATP hydrolysis displacing the RNA through the channel and essentially creating a “power stroke”. The arginine finger motifs also show a slight rearrangement moving approximately 1 and 2 Å, respectively, toward the nucleotide-binding site in an adjacent subunit. The RNA appears to stimulate ATP hydrolysis. Amino acid residues involved in ssRNA binding are in a cleft between two P4 monomers.

The specifics of the interaction of the P4 hexameric motor with the P1 in both the procapsid and the nucleocapsid in regard to occupancy, points of contact, and stability are not entirely worked out; however significant starting data were obtained that can be used as a guide for future study. In studies by Nemencek et al. [61], the P4 hexamer was seen to occupy randomly and incompletely on average 5 out of 12 PC recessed vertices of phi6. However, it appears that in the final assembled and filled NC, the P4 hexamer occupancy is complete (12 hexameric units). In our own initial research based on the density measurements of the PC by tomographic reconstructions and subtomogram averaging, the incomplete hexamer occupancy was determined at an even lower number which averaged to approximately 2 to 3 hexameric rings per PC particle [76]. The low occupancy of P4 motivates questions as to how the hexamer is transiently bound in the unfilled PC and how and why it is stably maintained at full occupancy once the PC is RNA-filled and incorporated in the NC? Future studies must examine the precise interactions between P4 and P8 compared to that between P4 and P1. The regression analysis of single-particle reconstructions demonstrated that decreased presence of P8 on an RNA-filled and P1 expanded NC was associated with a decreased presence of the P4 hexamer. Therefore, the conclusion (perhaps self-evident) revealed that contact regions or points between P8 and P4 are responsible for the stability of the P4 hexamer and the complete hexamer occupancy at viral maturation. The precise number of contact points (or density bridges as seen by cryo-electron tomography) is dependent on the reconstruction threshold value utilized; however, in these initial measurements, five contact points were noted (Figure 10a). However, in spite of only having preliminary observations, a dynamic description begins to emerge from extant studies.

As reported by Sun et al. [60], phi6 PC assembly requires P4 at every vortex, an observation in seeming contrast to the observed low P4 occupancy of the unfilled PC. Subtomogram averaging revealed a tenuous contact between P1 and P4 at or very near the vertices in the concave region of the portals of the unexpanded PCs (Figure 10b). The same idea was confirmed by Lisal et al.’s study, where the rates of hydrogen–deuterium exchange of the phi12 P4 (compared both in solution and bound to the P1 framework of PC) showed that P4 associates with the PC at its C-terminal domain [71,77]. The connection between P4 and P8 is distinct from the connection between P4 and P1. The P1–P4 connection is present in both the unexpanded PC and the mature NC. In the NC, connections between the P8 lattice with P4 likely have a greater affinity than the P1–P4 connections. Supporting this notion, a reconstruction of the portal with relaxed symmetry shows asymmetric binding of the P4 hexamer to P1 at the fivefold vertex, and the C-terminal region of P4, modeled as disordered in the crystal structure was shown to be required for binding to the PC. The cryo-electron tomography data that were obtained needs to be collected using up-to-date techniques that allow improved and higher resolution reconstruction images. However, a conceptual model can be proposed that accounts for asymmetry and a transient restriction on portals and allow for the observed order of packaging of ssRNA into the φ6 PC. In the PC assembly pathway, P4 appears to be shed and transiently attached, perhaps facilitating a limited number of portals used during packaging. In further support of the model, a mutant S250Q produced particles with a greatly reduced (10 to 20%) amount of P4 per particle yet are capable of packaging ssRNA, suggesting that only one portal is active as a packaging entrance. Since packaging of the ssRNA is known to be sequential in the order s, m, and l, this observation also supports a model where only a single packaging portal is active. Future experimentation might be designed to measure if the P4 hexamer has a quantifiable half-life at the PC 5-fold axis. Only when packaging and dsRNA replication has completed the assembly of the P8, the lattice stabilizes of all the symmetrically mismatched P4 hexamers at each of the twelve pentameric faces. In a subsequent cell infection, the NC is capable of ssRNA transcription from all the twelve P4 hexamers. However, the apparent absence of the P8 lattice in phi8 appears to contradict this model and again shows that additional study is required.

## 8. The Precise Function and Positioning of Protein P7

Controversy remains regarding both the precise location of protein P7 and its function. While sometimes referred to as an “accessary” protein, this term is misleading in that this component is present in all the cystovirus PCs and facilitates assembly dynamics [31,78]. However, P7 remains the least characterized of the PC proteins, with the precise function still not determined. However, tantalizing clues have been suggestive of the role the protein plays in viral assembly and stability.

P7 is required for efficient PC assembly and transcription [55] and RNA packaging [30,79]. In phi6, P7 has a molecular mass of 17,168 Da. Based on L. Day and L. Mundich study [78] and symmetry of the virion, the phi6 virion can potentially contain 60 copies of P7 (three copies at each of the 20 three-fold symmetry axes); however, different research groups reported a variable number of P7s in recombinant PC particles. Sun et al. [60] noted that the same amount of P7 is in recombinant PC particles as in the complete virion. Our lab estimated approximately 20 copies of P7 protein per PC particle [32], while Nemecek et al. [80] observed even less occupancy for P7, at only 12 copies in a complete PC. The occupancy of P7 in mature viruses was not determined through direct observation and may differ from recombinant PC particles. There is evidence that P7 forms an elongated dimer in solution [55], but in both the PC and NC, P7 is seen to exist as a monomer (Figure 11). Poranen et al. [33] observed that an excess amount from the stoichiometry of P7 accelerated assembly of P1 in vitro, indicating that P7 may stabilize P1. The study of self-assembly of PC particles shows potential competition between P2 and P7 proteins [80]. In the presence of an excess of P7, the number of P2 copies decreases from 12 to 6 copies, whereas the number of P7 copies increases from 36 to 60 copies per particle [60]. Our group proposed that P7 is located “inside” the PC core near the inner three-fold axes, and P7 stabilizes P2 in a position close to the three-fold axis [32]. We noted that using antibody reagents, during phi6 NC transcription, the P7 becomes accessible to the antibody, implying the loosen P1 matrix and P8 shell during the transcription [81]. The P7 protein has very low similarity between species by sequence, but structurally they are very similar. The structure of P7 from the related cystovirus phi12 was determined by X-ray crystallography for the first 129 aa [82] (Figure 11c). The protein is seen to be composed of two distinct domains in which the N-terminal core region (1–129) of P7 forms a homodimeric α/β-fold, with structural similarities with BRCA1 C-terminal domains implicated in multiple protein–protein interactions in DNA repair proteins. The nonstructural C-terminal tail consisting of approximately 30 to 40 amino acids is less ordered structurally, and nuclear magnetic resonance (NMR) analysis strongly suggested that it might be capable of interaction with the viral RNA and the RdRP [83]. Specifically, the acidic cluster on the P7 C-terminus induced chemical shift perturbations in the NMR spectra of P2 on domains in proximity to the RNA template tunnel. The mechanistic proposal was presented explaining that this acidic cluster interaction is similar to ssRNA binding to P2, providing a “stabilization” in the positively charged tunnel hence a regulatory function on the RdRP. However, the interactions were observed only in solution, and future work needs to determine if the phenomenon occurs in the context of an assembled PC and the entire virion.

Preliminary data to this end utilized different maps generated from single-particle reconstructions of the PC. In phi6, P7 is located very near the inner three-fold axis of symmetry in the unpackaged and unexpanded PC [32]. In this imaging study, it was noted that P7 indeed interacts with P2, and it is proposed that the P7 protein stabilizes the location of P2 near the inner three-fold axis. In a sense, the P7 appeared to provide a mechanical function in that it “holds” the P2 RdRP near the three-fold axis of the unfilled PC (Figure 11a).

In the PC mutants lacking P7, P2 organized in a less ordered fashion and drifted from the inner 3-fold axis, and in the PC mutant lacking P2, P7 occupancy is substantially reduced. Although measurements and quantitation based on electron density from image reconstruction can only approximate occupancy levels, the data are consistent with the solution NMR analysis described above. The P7–P2 interaction could inhibit P1 shell movement prior to RNA packaging that might stabilize an unexpanded PC. The notion is supported by hypotheses put forth by Butcher et al. [24] and Eryilmaz et al. [82] that P7 acts as a hinge for P1 expansion during RNA packaging. The location of P7 proteins after the PC expansion in NC remains controversial and requires future study. In summary, the data are strongly suggestive that P7 is not a static component and possesses a significant genome packaging role in which RNA is guided to the P2 RdRP, which then decouples from P7 in the expanded PC.

## 9. Final Maturation of the Nucleocapsid and Virus Particle

The viral nucleocapsid maturation is completed when the P8 lattice is added to the filled and expanded PC. P8 trimers are organized in an incomplete icosahedral lattice. Phi12 and phi6 have comparable numbers of the trimers: 200 or 600 monomeric proteins, but the phi8 virus has a loosely organized P8 layer, and the number of P8 barely equals 60 [84]. The arrangement of the trimers of P8 determines the architecture of the NC. The NC matrix contains two types of vacant regions (termed “holes”) that are designated I and II and composed of 60 of each type determined by the arrangement of the P8 trimers.

In phi6, there are four classes of interdigitating P8 trimers termed Q, R, S, and T, with Q closest to the five-fold axis of symmetry (see Figure 12A,B) [22]. The five-fold axis formed by the P8 layer is fully occupied by the P4 hexamer, and as noted above, the precise interactions between these two NC elements are still open to question. The phi6 lattice contains two states of interdigitating P8 trimers that are designated as either open or closed. The trimers can shift between the two states in order to swap domains, presumably to stabilize the viral outer shell.

The final step in cystovirus assembly is the acquisition of the bi-lipid outer membrane. Nonstructural protein P12 can be involved in viral envelope biogenesis as a lipid transporter, membrane stabilizer, or/and protease inhibitor, but its precise function remains undefined. Lyytinen et al. [85] indicated that (at least in *E. coli*) the protein P12 co-expressed with major membrane protein P9 promoted the formation of stable incorporation of P9 into membrane vesicles and had proteolytic inhibiting activity.

Single-particle cryo-EM reconstruction of the phi12 virus by Wei et al. identified a feature in the entire particle extending toward the lipid layer and loosely associated with the NC [20]. The feature was postulated to be the murein peptidase P5, and the reconstruction analysis strongly suggested that it could be symmetrically arranged within the viral particle (see Figure 12C,D). This model allows 60 P5 protein units per virion, which is considerably less than the approximately 90 P5 units estimated by isotopic labeling [78]. The densities that are tentatively assigned to P5 appeared to be located at a position, which is the equivalent of the phi6 NC type III holes. [22]. In the phi6 virus, the type III holes are empty, and it is possible that the holes might provide sites to accommodate proteins in conformations specific to the replication and assembly mechanisms for a particular cystovirus species. The P5/P11 position remains an open question with significant evolutionary and structural implications.

## 10. Conclusions and Suggestions for Additional Areas of Investigation

The viral genome packaging mechanisms and associated dynamic alterations of the cystovirus capsid were delineated to a great extent by the judicious use of genetic analysis in conjunction with structural studies. However, as described in the review, questions remain, and additional studies are needed to provide the answers fully. These are summarized and listed in the seven points that follow as a guide to encourage further studies. No particular order of importance is implied.

The temporal regulation of transcription is incompletely described as the discovery by Qiao et al. [47,48] that an ortholog of the cellular host factor, YajQ, is crucial to the function and synthesis of the l transcript. The protein appears to bind the P1 and affect the RNA pol P2 (perhaps by a structural alteration), but the precise mechanism remains undefined.

The precise role of protein P7 in capsid assembly and RNA packaging remains unknown. Indeed, it remains unclear if the protein exists as a dimer or monomer when assembled within the capsid. The data from our own research efforts [28,32,81] strongly suggested a structural role for a monomeric protein as a stabilization element for the P2 polymerase prior to and during transcript packaging. The absence of P7 PC particles showed a random distribution of P2 monomers throughout the particle in support of this notion. The alternative idea that P2 and P7 are mutually exclusive at the inner PC axis contradicts the model we proposed, and resolution of the debate is warranted.

The data describing the partial occupancy of the P4 hexamer on the unpackaged PC were obtained using recombinant particles in *E. coli*. There was no P8 synthesized in the cells, and the NC could not complete maturation. However, at maturation, all 12 NC pentameric faces are occupied with P4 hexamers forming the symmetric mismatch conformation. Only preliminary data exist showing the nature of the binding of P4 to the P1 frame and P8 lattice. The precise sites of interaction need to be determined at the near-atomic level, a goal within reach using cryo-electron microscopy techniques. The understanding of the binding sites of these three proteins will further the understanding of the viral RNA portal and the dynamic alterations it undergoes during the phases of the viral replication cycle.

Temporal regulation of the P2 polymerase RNA synthesis function is not yet understood. During the packaging phase of viral replication, the current data indicate only one P4 hexamer energetically inserts viral transcripts in the specific order s, m, and l. P2 only activates when the PC is completely filled, synthesizing the minus RNA strands to complete the dsRNA segments. The switch that activates the polymerase has not been described, and furthermore, the implication is that only one P2 pol is active at the packaging.

The viral envelope acquisition facilitated by the nonstructural P12 protein needs to be determined. The interaction of P12 with the host cell lipids with delivery to the NC surface lattice for envelope biogenesis remains little explored.

The data from the phi12 complete virus and PC reconstructions suggested a location for the murein peptidase, P5, in the type III holes of the P8 lattice. The observation is interesting not only from a structural point of view but at an evolutionary level as well. The suggestion that the holes are available for occupancy by specific proteins selected to best adapt a particular virus to a given environment answers or at least develops questions related to species fitness.

Finally, the entire P8 lattice is absent in bacteriophage phi8, yet the virus assembles replicates as efficiently as the other cystoviruses. This, as in point 6, has both structural as well evolutionary implications. It would be of interest to isolate a comparable P8 negative virus particle and correlate the species with any unique environmental conditions.

## Figures and Tables

**Figure 1 ijms-23-02677-f001:**
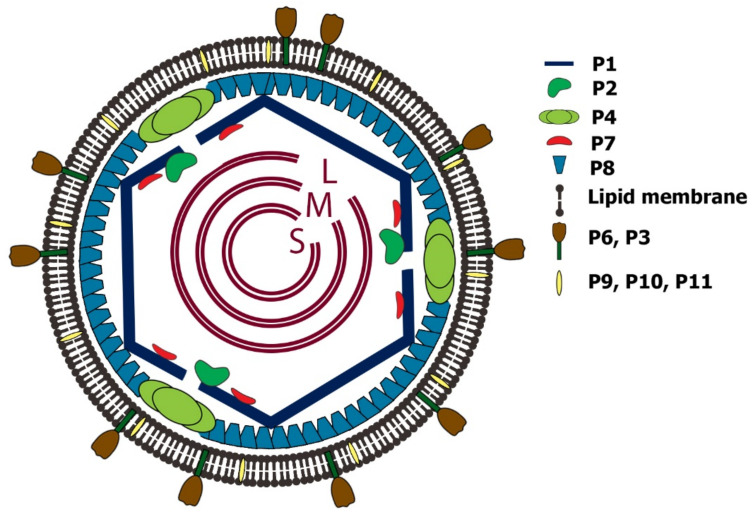
Schematic depiction of phi6 virion. The phi6 virus is comprised of three layers: inner layer or procapsid, the P8 matrix layer, and an outer envelope. The procapsid PC includes the dodecahedral T = 1 shell, which is composed of 120 copies of P1 protein (60 non-symmetrical dimers of P1A/P1B). Inside the shell, there are the three low-symmetry dsRNA segments and P2 and P7 proteins organized at the 5-fold axis. The P4 proteins form a hexameric ring around the 5-fold axis portal; the icosahedral T = 13 NC includes the PC and matrix layer composed of P8 proteins. The complete virion has an envelope from the cellular bi-lipid membrane that contains randomly distributed P3, P6, P9, P10, and P5/P11 proteins.

**Figure 2 ijms-23-02677-f002:**
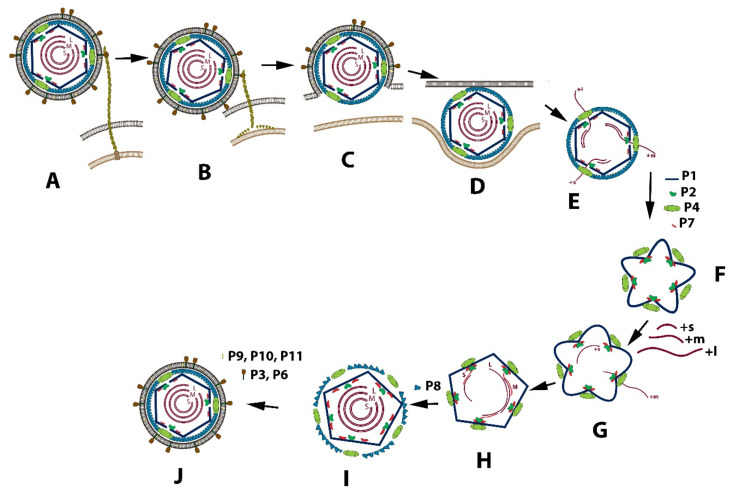
Schematic depiction of the replication cycle of the phi6 virus. (**A**) P3 binding protein attaches to type IV pili; (**B**) the pili retract, and the virion repositions in close proximity to the bacterial membrane; (**C**) the viral envelope fuses with the bacterial outer membrane; (**D**) phi6 NC enters into intracellular space; (**E**) the viral core releases into cytoplasm, the P8 layer disassembles, and transcription initiates; (**F**) after expression of major structural proteins P1, P2, P4, and P7, the proteins assemble into the procapsid PC; (**G**) RNA packaging initiates in the following order: +s, +m, +l, and PC starts to expand; (**H**) + RNA strand replicates to the ds-RNA genome; (**I**) the P8 matrix loosely assembles around PC; (**J**) the cell-derived bi-lipid membrane incorporates into the virion structure, followed by cell lysis and virion release.

**Figure 3 ijms-23-02677-f003:**
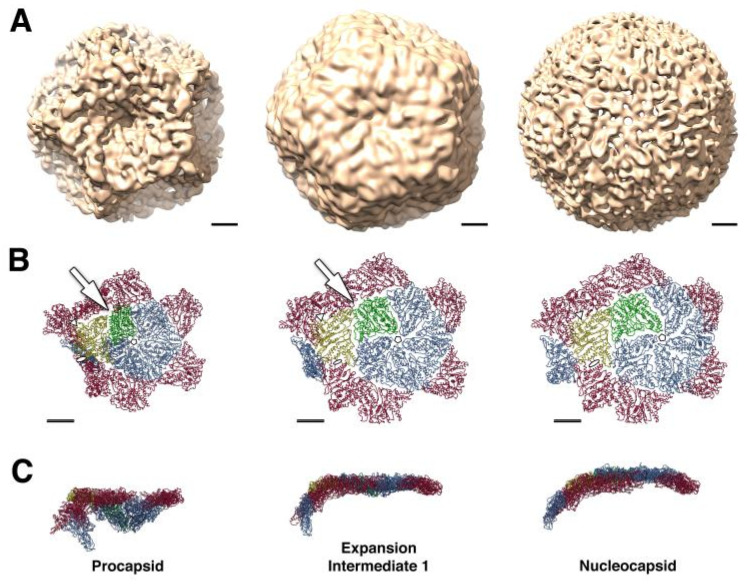
Stepwise procapsid expansion from collapsed procapsid (left images) through intermediate expansion (central images) to the nucleocapsid (right images). (**A**) CryoEM reconstructions of three conformational states of the phi6 capsid at 16 Å resolution. (**B**,**C**) Representation of part of the capsid around the 5-fold axis. A pentamer of P1A subunits (blue, green) surrounded by P1B subunits (red, yellow), viewed from above (**B**) and from the side (**C**). Initial to intermediate expansion achieved by rotation of P1B subunits (yellow) around an axis connecting the 3-fold icosahedral axes (indicated by white triangles on panel **B**). This rotation stabilizes the P1B/P1B interface at the 2-fold axis and seals gaps between P1A and P1B subunits near the 3-fold axis (indicated by white arrow). Outward movement of P1A subunits provides a future expansion to the intermediate 2nd stage (not shown) and finally to NC. The expansion is also accompanied by local conformational changes in the P1A subunits that correlate with increased outward curvature at the 5-fold axis. Scale bar: 100 Å. Reproduced from Nemecek et al. [21].

**Figure 4 ijms-23-02677-f004:**
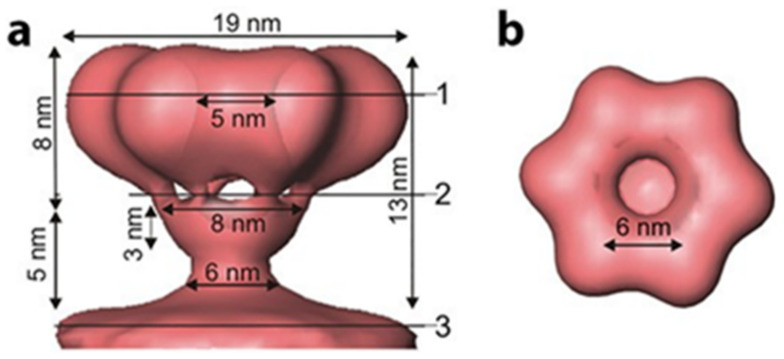
The 3D reconstruction of P3 phi12 attachment protein complex. The complex consists of six globular copies of P3a and three copies of P3c. Data reproduced from Leo-Macias, A. et al., 2011 [38]. (**a**) represents the side view of the attachment complex and (**b**) is the top view.

**Figure 5 ijms-23-02677-f005:**
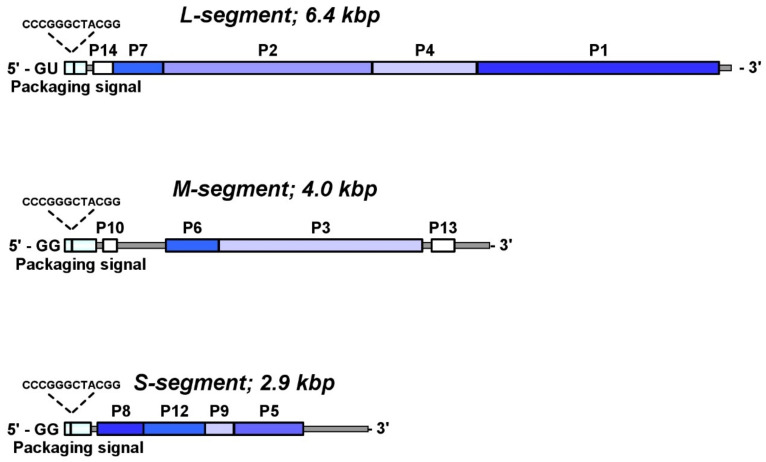
Genome organization of the phi6 virus shown according to the three dsRNA segments encode the viral proteins. The genes encoded in the structural proteins are indicated by blue shades. Nonstructural genes are indicated by white, and the packaging signal is shown in light blue. The first conserved 18 nt at 5′ end of the segment did not show except the two initial nucleotides: GG for S and M segments and GU for L segment.

**Figure 6 ijms-23-02677-f006:**
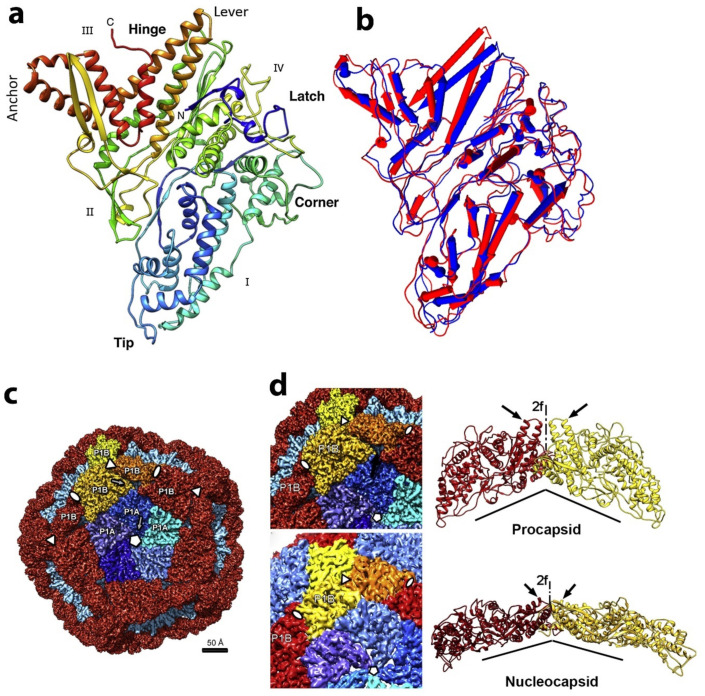
P1 protein structure and P1 shell organization in the PC and the NC. (**a**)—Trapezoidal P1 subunit (rainbow-colored from blue at the N terminus to red at the C terminus): the four edges are labeled as I–IV. The rotated long helix-turn-helix (in gold) forms a “lever” that can shift during the maturation and expansion of the procapsid. The five P1A “tips” arranged around the inner part of axial channel of the pentamer. The “corner” of one pentamer-formed subunit P1A fits against the “anchor” of a neighboring subunit P1B. Reproduced from Nemecek et al., 2013 [21]. (**b**) Superposition of P1A (red) and P1B (blue) subunits. The helixes are depicted as rods. The major structural differences are located at the “hinge” region, where the helixes rotated at ~18°. (**c**) Cryo-electron reconstruction and segmentation of phi6 PC. The outer surface viewed along a 5-fold axis. The 12 inverted 5-fold vertices are occupied by P1A pentamers (different shades of blue) fit in a dodecahedral frame of 60 P1B subunits (red, with the exception of the three yellow-colored subunits around one 3-fold axis). (**d**) P1B subunits (red and yellow) are tightly connected to P1A subunits (blues) of the inverted vertices in the procapsid (top images). This shell contains gaps between the P1B and P1A subunits clearly seen in ribbon representation on the side of the 2-fold axis (arrows). After maturation and expansion to the nucleocapsid (bottom images), the flat P1B subunits are rotated so that the planes of these molecules perfectly dock to the tangential plane of the shell, leaving no significant gaps between neighboring P1B. The subunit planes of P1B are almost perpendicular to each other in the procapsid, and corresponding representation of two P1B subunits in the nucleocapsid is almost co-planar. (The views are rotated around the 2-fold axis so that the dihedral angles appear considerably larger) Reproduced from Nemecek et al., 2013 [21].

**Figure 7 ijms-23-02677-f007:**
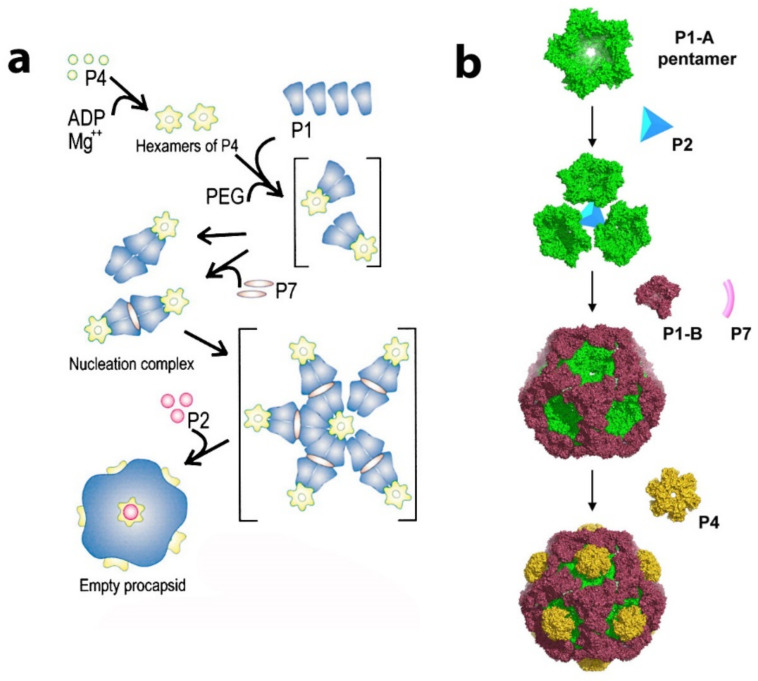
Proposed models of PC assembly pathway: (**a**) formation of tetrameric P1 units as a nucleation center. Reproduced from Poranen et al., 2001 [31]. (**b**) P1A pentamer as a building block for phi8 PC. Reproduced from El Omari et al., 2013 [53].

**Figure 8 ijms-23-02677-f008:**
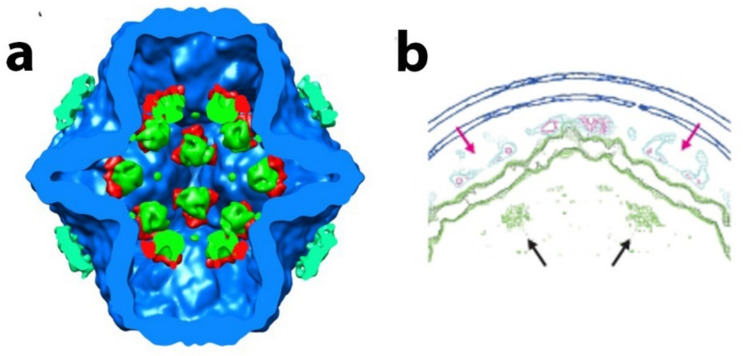
Empty and filled PC reconstructions. (**a**) Isosurface rendering of collapsed phi6 PC based on difference map segmentation of the PC into P1 (blue), P2 (green), P4 (cyan), and P7 (red). Image reproduced from G. Katz et al. [32] (**b**) Mesh-surface representation of the cross-section projection of rendering of two 5-fold axes from the entire phi12 virus particle showing the P2 RdRP directly below the P4 turret. The P2 densities are shown in green, and location of P2 is indicated by black arrows. The P4 NTPases are cyan in color and the portal shown by red arrows. Blue is the P8/P4 layer. Image reproduced from H. Wei et al. [20].

**Figure 9 ijms-23-02677-f009:**
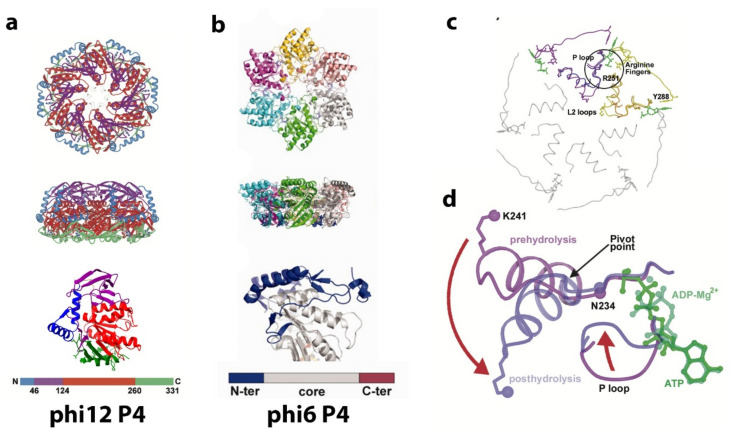
Structure of the P4 NTPase. (**a**) The P4 hexamer of phi12 is shown in terms of its secondary structural elements and solvent accessible surface in top and side views. The secondary structural elements are colored according to the bar where different colors distinguish subdomains or segments of the P4 monomer: the N-terminal safety pin motif (blue) holding the six monomers together, all beta domain (purple), the conserved RecA-like ATP binding domain (red), and the antiparallel β strands and C-terminal helix that is close to the PC (green). The image was reproduced from Mancini et al. [63]. The bottom panel depicts the P4 monomer in the “side” orientation. The color code corresponds to the hexameric form. The structural data is taken from 1W44 PDB. (**b**) The P4 hexamer of phi6 is shown as a top and side views. Each P4 monomer is depicted in different colors. The bottom panels show structures of monomeric P4 in the upper orientation of the monomer corresponding to the one depicted in cyan in top panels; the core domain is colored in gray, the N-terminal domain in blue, and the C-terminal domain in red. Nucleotides, if present, are depicted as sticks with carbon, oxygen, nitrogen, and phosphorus atoms colored in yellow, red, blue, and orange, respectively. Dotted lines represent the disordered region of the proteins. Image reproduced from El Omari et al. [64]. (**c**) Coil representation of the major secondary structure elements for the entire P4 hexamer. The neighboring subunits are shown in magenta and yellow. The superimposed conformations of AMPcPP-Mg^2+^ to ADP-Mg^2+^ are shown in purple and violet for first subunit and light to dark yellow for the neighboring subunit. The encircled area is shown in detail in (**d**). (**d**) Details of the nucleotide-binding domain and corresponding P loop and L2 loop in the AMPcPP “A” conformations superimposed on the ADP-Mg^2+^ (“P”-like) conformations colored as in (**c**). The red arrows indicate the conformational changes which the α6 helix-L2 loop and P loop undergo following hydrolysis of bound ATP Date reproduced from Mancini et al. [63].

**Figure 10 ijms-23-02677-f010:**
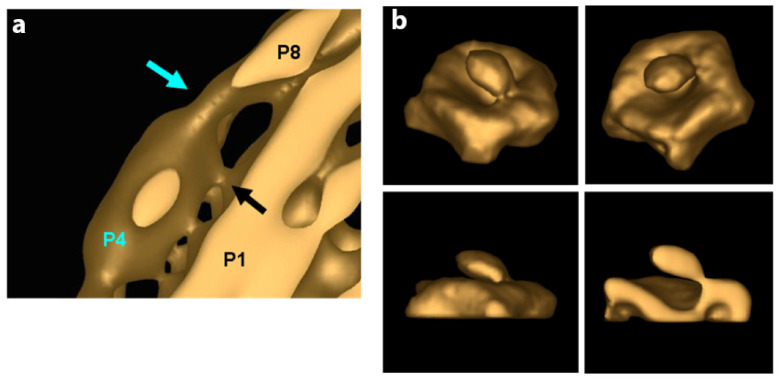
Low-resolution cryo-EM tomogram reconstructions of the occupied portal with a P4 hexamer centrally located. (**a**) Blue arrows show the density where the P4 hexamer contacts the P8 lattice. The black arrow shows the densities where the P4 hexamer contacts the P1 framework. (**b**) The density loosely attaching the P4 hexamer to the P1 framework is shown in three angles and a cutaway view (in the lower right panel). Reproduced from Ph.D. thesis by J. Carpino, “Structure and Function in Bacteriophage Phi6”, 2014 [76].

**Figure 11 ijms-23-02677-f011:**
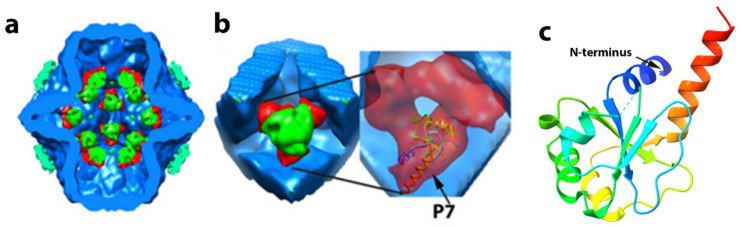
Proposed P7 protein positioning in phi6 the unexpanded PC and phi12 P7 crystallographic structure as a ribbon diagram. (**a**,**b**) The phi6 P7 protein localizes within the PC based on a cryo-image difference map created from 3D reconstruction of PCs mutants lacking different structural proteins (lacking P2 or P7) and complete PC particles (have P1, P2, P4, and P7 proteins). The central cut of the PC complex shows the localization P2–P7 complexes at the 5-fold axis proximity. P2 protein is shown in green, P7 is in red. P1 density is blue, and P4 hexameric rings are in cyan. Magnified section (**b**) shows P2 surrounded by three P7 monomers and an insert showing docked atomic model of P7 based on X-ray resolved structural data for phi12 P7. Data reproduced from G. Katz et al. [32]. (**c**) Crystal structure of N-terminus domain of phi12 P7. PDB data 1Q82. Nonstructural C-terminus are not shown.

**Figure 12 ijms-23-02677-f012:**
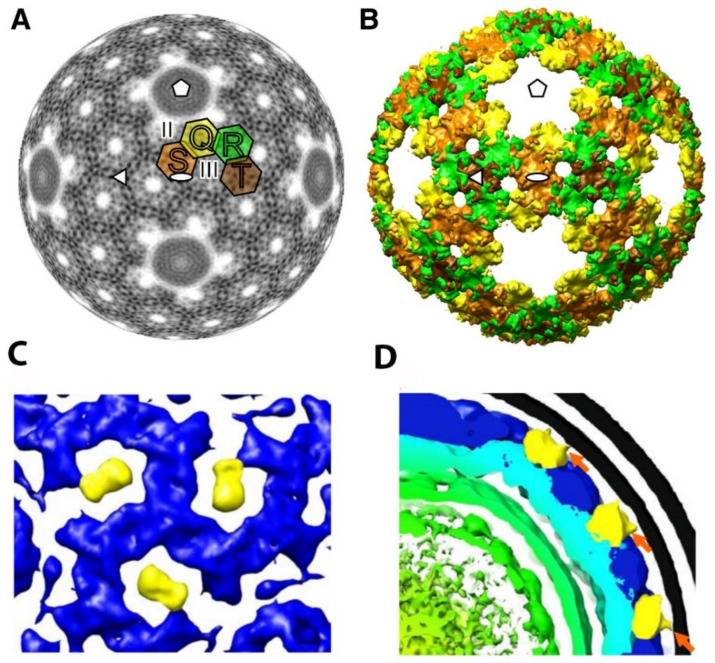
The phi6 P8 layer organization and P5/P11 arrangement. (**A**) Electron density layer (at a radius of 263Å) from the phi6 NC reconstruction. The four unique P8 trimers are labeled with Q (yellow), R (green), S (light brown), and T (brown). (**B**) Isosurface rendering of the phi6 P8 layer using the threshold corresponding to the calculated protein mass (1.2 above the mean electronic density). Reproduced from J. Huiskonen et al. [22]. (**C**) Enlarged area of a top view along the 3-fold axis reveals densities surrounding the axis. The densities tentatively assigned to P5/P11 (yellow). These densities are significantly less dense than the P8 proteins (blue) in the NC layer. (**D**) Low threshold assignment for lipid bi-layer (black lines) and P8 layer (blue) reveals the connections between the bilayer and the novel densities tentatively assigned to P5 (orange arrows). Reproduced from H. Wei et al. [19].

**Table 1 ijms-23-02677-t001:** Major structural proteins of *Cystovirus* phi6, its molar weight, number of copies per virion, and basic cell functions.

Protein	Molar Weight, KDa ^1^	Copies per Virion ^2^	Function
P1	93.0	120 or 60 dimers	Major structural protein
P2	74.8	12	RNA-directed RNA polymerase
P4	35.0	72 (12 hexamers)	NTPase packaging motor, transcription
P7	17.2	60	Assembly cofactor; packaging cofactor
P5/P11	24.0		Lytic endopeptidase; murein peptidase
P8	15.8	600 (200 trimers); 720	Shell
P6	21.0		Integral membrane protein
P3	84.0		Binding protein
P9	9.5		Major envelope protein;
P10	6.0		Putative holin protein

^1^ Calculated form cDNA sequence. ^2^ Estimated copy number, based on SDS-PAGE gel and cryo-electron microscopy data.

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
