# Peer review of "RNA Packaging in the Cystovirus Bacteriophages: Dynamic Interactions during Capsid Maturation"

_ijms, 2022, doi:10.3390/ijms23052677_

Round 1

Reviewer 1 Report

The manuscript entitled RNA Packaging in the Cystovirus Bacteriophages: Dynamic Interactions During Capsid Maturation describes the developments of the RNA packaging and replication mechanisms, focusing the attention on the following five proteins: P1, P2, P4, P7 and P8. Moreover, the authors illustrated all open questions on this topic that could be a starting point for further analysis.

The manuscript is interesting and well structured. There are just a few oversights that should be fixed:

-line 47, gran-negative should be changed for gram-negative;

-lines 192/193, there is the same concept as in the previous sentence (While in most members of the Cystoviridae the order of the L segment genes is 7,2,4,and 1);

-line 208, a crossed-out sentence is present, that is (represented by phi6);

- line 218, change of for or;

-legend of figure 4, please explain what a and b are.

Author Response

Dear Editors and Reviewer;

Thank you for all your valuable notes. We agree with all oversights and fixed it. The detailed point-by-point corrections are below:

-line 47, gran-negative should be changed for gram-negative; - corrected

-lines 192/193, there is the same concept as in the previous sentence (While in most members of the Cystoviridae the order of the L segment genes is 7,2,4,and 1); Corrected to : The genes encoding the structural proteins of PC, are located on the L segment. While in most members of the Cystoviridae the order of the L segment genes is 7, 2, 4, and 1….

-line 208, a crossed-out sentence is present, that is (represented by phi6); - corrected

- line 218, change of for or; - corrected

-legend of figure 4, please explain what a and b are. - Description of A and B added : A – represents the side view of the attachment complex and B is the top view.

Reviewer 2 Report

This is an important and interesting review article on the cystoviral genome packaging mechanisms. Cystoviruses are specific bacteriophages which can serve as models of dsRNA viruses, thus, they are very intersting research subjects. The paper is generally well-presented and deserves publication. However, minor (though significant) revision is required. Specific points are listed below.

  1. Although the language is generally good, it looks like if the paper was written in hurry... Especially, many typographical errors should be corrected, and the authors seem to ignore interpunction. Please, re-check the text throughout the paper correct those simple errors.
  2. Genetic, biochemical and microbiological nomenclature is confusing in some places. Please, remember that names of genes should be written in italic font while names of proteins should be written using regular font. Moreover, names of species and genera should be written according to biological taxonomy rules (for example, line 54: is should be Pseudomonas sp. B314 instead of Pseudomonas sp. B341). Please, re-check the text nad follow the genetic and microbiological nomenclature rules.
  3. Table 1 is not cited in the text. Please, cite it.
  4. In my opinon, it would be better to present the phage genome organization (current chapter 4) and RNA transcription (current chapter 5) just after presentation of general information about Cystoviridae. This is, however, optional, and I leave the decision to the authors.
  5. Please, be consistent with names of bacteriophages. For example, two names are used for the same virus: Phi6 (e.g. line 45) and Φ6 (line 56). Both names are correct, but the authors should use only one of them throughout the text, to avoid confusion.
  6. Line 47: Replace "gran-negative" with "Gram-negative" (the name is after surname of a researcher, thus, it should be "Gram").
  7. Line 395: Replace semicolon with a period after "review".
  8. Chapter 10: Provide references also in this chapter, despite it describes conclusions. For example, "discovery by Qiao et al." (line 628) is confusing if not followed by reference. The same for "The data from our own research" (line 633) - please provide appropriate reference(s). This is valid for the whole chapter.

Author Response

Dear Editors and Reviewer;

Thank you for all your valuable notes. It helps to make our review paper better. We agreed with all oversights and fixed it. The detailed point-by-point corrections are below in italics:

  1. Although the language is generally good, it looks like if the paper was written in hurry... Especially, many typographical errors should be corrected, and the authors seem to ignore interpunction. Please, re-check the text throughout the paper correct those simple errors. – we have checked the text and corrected all typos and punctuational errors.
  2. Genetic, biochemical and microbiological nomenclature is confusing in some places. Please, remember that names of genes should be written in italic font while names of proteins should be written using regular font. Moreover, names of species and genera should be written according to biological taxonomy rules (for example, line 54: is should be Pseudomonas sp. B314 instead of Pseudomonas sp. B341). Please, re-check the text nad follow the genetic and microbiological nomenclature rules. - corrected
  1. Table 1 is not cited in the text. Please, cite it. – now cited in lines 76-77: The functions of basic structural proteins, its molecular weight and the number of copies per virion are presented in Table 1.

4. In my opinon, it would be better to present the phage genome organization (current chapter 4) and RNA transcription (current chapter 5) just after presentation of general information about Cystoviridae. This is, however, optional, and I leave the decision to the authors. – Thank you for your opinion, we took it into the consideration, but we feel the current order is correct.

5. Please, be consistent with names of bacteriophages. For example, two names are used for the same virus: Phi6 (e.g. line 45) and Φ6 (line 56). Both names are correct, but the authors should use only one of them throughout the text, to avoid confusion. – corrected.

6. Line 47: Replace "gran-negative" with "Gram-negative" (the name is after surname of a researcher, thus, it should be "Gram"). - corrected

7. Line 395: Replace semicolon with a period after "review". - corrected

8. Chapter 10: Provide references also in this chapter, despite it describes conclusions. For example, "discovery by Qiao et al." (line 628) is confusing if not followed by reference. The same for "The data from our own research" (line 633) - please provide appropriate reference(s). This is valid for the whole chapter. – the references are included into the text. The chapter 10 provides conclusions and the brief review of questionable and incompletely described recent literature and we included the references only to sentences which can be confusing and required the citations.